# Age-dependent seroprevalence of dengue and chikungunya: inference from a cross-sectional analysis in Esmeraldas Province in coastal Ecuador

Irina Chis Ster [ORCID],[1] Alejandro Rodriguez,[2] Natalia Cristina Romero,[2] Andrea Lopez,[2] Martha Chico,[2] Joel Montgomery,[3] Philip Cooper[1,2]

► Prepublication history and additional materials for this paper is available online. To view these files, please visit the journal online (http://dx.doi.org/10.1136/bmjopen-2020-040735).

[1]Institute of Infection and Immunity, St George's University of London, London, UK
[2]International University of Ecuador, Quito, Ecuador
[3]CDC, Atlanta, Georgia, USA

**Correspondence to**
Dr Irina Chis Ster;
ichisste@sgul.ac.uk

## ABSTRACT

**Objectives** There are few population-based estimates for prevalence of past exposure to dengue and chikungunya viruses despite common epidemiological features. Here, we have developed a novel statistical method to study patterns of age-dependent prevalence of immunity in a population following exposures to two viruses which share similar epidemiological features including mode of transmission and induction of long-lasting immunity. This statistical technique accounted for sociodemographic characteristics associated with individuals and households.

**Settings** The data consist of a representative sample from an ongoing longitudinal birth cohort set-up in a tropical district in coastal Ecuador (Esmeraldas).

**Participants** We collected data and blood samples from 319 individuals belonging to 152 households following epidemics of the infections in 2015 in Latin America.

**Primary outcome** Plasma was tested for the presence of specific IgG antibodies to dengue and chikungunya viruses by commercial ELISA and defined a bivariate binary outcome indicating individuals' past exposure status to dengue and chikungunya (ie, presence/absence of IgG antibodies to dengue or chikungunya or both).

**Results** Dengue seroprevalence increased rapidly with age reaching 97% (95% credible interval (CrI): 93%–99%) by 60 years. Chikungunya seroprevalence peaked at 42% (95% CrI: 18%–66%) around 9 years of age and averaged 27% (95% CrI: 8.7%–51.6%) for all ages. Rural areas were more likely to be associated with dengue-only exposure while urban areas and shorter distance to the nearest household were associated with exposures to both. Women living in urban settings were more likely to be chikungunya seropositive while rural men were more likely to be dengue seropositive.

**Conclusion** Dengue seroprevalence was strongly age dependent consistent with endemic exposure while that of chikungunya peaked in childhood consistent with the recent emergence of the virus in the study area. Our findings will inform control strategies for the two arboviruses in Ecuador including recommendations by the WHO on dengue vaccination.

## INTRODUCTION

Dengue virus (DENV) and chikungunya virus (CHIKV), arboviruses belonging to *Flaviviridae*

### Strengths and limitations of this study

► This is the first serological study presenting population-based estimates on age-dependent prevalence of dengue and chikungunya in a region of Ecuador.
► The paper describes a novel statistical method and associated OpenBUGS code.
► The estimates account for individual and household characteristics.
► The paper responds to recent WHO call for understanding local epidemiology of dengue prior to designing efficient control strategies.
► More research is needed to assess the impact of various control strategies in the community using these estimates and their uncertainties.

and *Togaviridae* families, respectively, are endemic in many tropical and subtropical regions where *Aedes* vectors thrive. Although the two viruses belong to different arbovirus families,[1] human infections are associated with similar clinical manifestations and are a major cause of morbidity.[2–4] Over the past 15 years, the number of dengue cases reported in Latin America has increased dramatically[5] while chikungunya, which emerged in the Caribbean region in 2013, has spread rapidly through the region[6] causing an estimated 2.9 million cases.[7–9]

Global estimates of the burden of dengue and chikungunya are based on aggregate numbers of reported cases and likely to be biased by under-reporting because of limited surveillance, high rates of asymptomatic infections and non-specific clinical presentation.[10] Asymptomatic dengue, for example, can inflate the total number of cases by up to 40-fold.[11–13]

Cross-sectional serological surveys can provide valuable information on arbovirus infection burdens and allow more accurate

determination of numbers exposed. Serological surveys, thus, have a number of advantages over clinically classified, self-reported cases or other non-objective clinical diagnoses by providing information that is less sensitive to recall, under-reporting and misdiagnosis due to asymptomatic manifestations of related infections.[6 12–14]

Ecuador is a middle-income country in Latin America where dengue has been endemic since the late 1980s[15]: dengue is thought to have re-emerged around 1988 following recolonisation with *Aedes aegypti* after a 30-year absence.[16 17] Chikungunya first appeared in the Caribbean region in 2013, reaching Ecuador in 2014 causing a large outbreak the following year.[17 18]

Dengue transmission dynamics has been estimated from aggregated data in a series of studies[19 20] with the latter suggesting that the picture provided by serology is very incomplete and limited when trying to characterise transmission across time and space both within and between countries and calling for local data analyses.

Most seroprevalence studies for DENV and CHIKV have examined exposures to each in isolation despite evidence[21] of co-circulation and shared risk factors.[6 22 23] There are no published serological surveys of dengue and chikungunya in Ecuador or studies of co-exposures to the two viruses from the Latin American region,[6] and limited information on the determinants of exposure risk. Such data are critical to estimate the burden of infection and inform public health strategies for control. The first licensed dengue vaccine (CYD-TDV) is now available in the Latin American region. Current WHO guidelines advise the implementation of cost-effective prevention strategies that consider local arbovirus epidemiology.[24] Recent studies indicated that targeting only seropositive individuals for vaccination might reduce global annual disease incidence by 20%–30%, with the greatest impact in high-transmission settings.[25] Hence, understanding past exposure or seropositivity patterns in affected populations is the key to implementing cost-effective control strategies.

In this study, we hypothesised that prevalence of past exposure to DENV and CHIKV in the population is associated with age, and individual and household risk factors. We estimated age-dependent seroprevalence of DENV, CHIKV and co-exposures with both arboviruses in a district in coastal Ecuador, and explored individual and household determinants of past exposure status.

## MATERIALS AND METHODS
### Population and study sample
The data consist of a representative sample of households including children (and other household members) recruited into the ECUAVIDA birth cohort between 2005 and 2009 in the town of Quininde, Esmeraldas Province.[26] The district is largely rural in a tropical region of coastal Ecuador with a population of 88 000 living below 150 m altitude with mean annual temperature of 30°C and relative humidity of 80%. Main sources of income

are cultivation of palm oil and tropical fruits, and timber extraction. Children (and all other members of the household to which they belonged) due to attend the cohort clinic in Quininde for routine follow-up visits between October 2015 and February 2016 were invited to participate in this study according to the dates of their scheduled attendance. This approach resulted in the selection of a representative sample of ECUAVIDA cohort households (see online supplemental table 1). As is frequent for such ad-hoc surveys, post-stratification weightings[27] using the census population distributions were used to adjust the estimates to population-generalisable values (details in supplemental information and weights shown in online supplemental table 2.

Venous blood was collected into heparinised tubes (Vaccutainer, BD Systems), centrifuged and plasma was stored at −20°C. Data on individual and household factors were collected from the child's mother by an investigator-administered questionnaire and the 2010 census population data were used to derive post-stratification weights such that the resulting estimates would represent those of the population.[27] Each household was visited and location determined by Global Positioning System (GPS) (Garmin eTrex, Kansas, USA).

### Laboratory testing
Plasma was analysed for specific IgG antibodies to DENV and CHIKV using commercial ELISA kits (Human anti-Dengue IgG and human anti-Chikungunya IgG ELISA kits, Abcam, Cambridge, UK) following manufacturer's instructions. Past exposure or seropositivity to DENV and CHIKV was defined by the presence of specific IgG antibodies.

### Statistical methods
The presence or absence of DENV and CHIKV antibodies define a bivariate binary outcome indicating past exposure to DENV only denoted by D+C−, to CHIK only denoted by D−C+, to both, D+C+, or to none. The key assumption of the model is that of long-lasting immunity[19] after infection with DENV and CHIKV.

We used two simultaneous multilevel logistic regression equations with shared random effects to model the presence/absence of antibodies for DENV or CHIKV or both. The basic idea of this model was developed initially in econometrics[28] and more recently in the context of two binary correlated outcomes indicating infections but using a distinct approach from the one used here.[29] Each equation modelled the presence of one arbovirus conditioned on the other, making the presence of DENV more or less likely in the presence of CHIKV (and vice versa). We explored the effects of age, sex, area of residence (urban vs rural), socioeconomic level (grouped as low, medium and high derived from tertiles of the first component of principal components analysis of seven socioeconomic variables as described in[30]), crowding (number of persons per sleeping room) and household dispersal (Euclidian distance to nearest household using GPS coordinates)

on our bivariate binary outcome (dependent variable) defined above. The simplest model started with investigating the effect of age on the outcome which provided answers to the main question of interest, that is, how the probabilities of past exposures to (D+C−, D−C+ and D+C+) varied with age in this population. The systematic trends with age in these probabilities are called age-dependent prevalence of past exposure to D+C−, D−C+ and D+C+. They are also predicted probabilities of past exposures. We then refined these findings by exploring the extent to which these overall associations between past exposure to D+C−, D−C+ and D+C+ and age are altered by the other variables—alone and in a multivariable model.

The model (see online supplemental material) also embedded post-stratification weights, allowing estimates to be generalisable to the population from which the sample was derived.[27] Hierarchical structure[31] of the data (individuals within families within wards) was represented by shared random effects, accounting for unmeasured epidemiological factors such as biting rates or climate conditions. Because of the assumption of long-life immunity, the age-dependent prevalence of past specific or co-exposures[14] can be also regarded as measures of immunity to these viruses in the population.

Bayesian inference was used for parameter estimation and to predict age-dependent probabilities of seropositivity and 95% credible intervals (CrIs). In the absence of any formal goodness-of-fit measure, predicted age-dependent prevalence was plotted against the observed proportions of positives within the same 5-year age group. Moreover, we also plotted the observed numbers of seropositive individuals across 5-year groupings and across geographical regions against expected numbers predicted by the model.

Detailed information on the statistical approach used with mathematical models, framework for statistical inference, estimated parameters and goodness of fit are provided as online supplemental material. Aggregated data at national and provincial levels for case reports of dengue/chikungunya from 2015 were retrieved from various sources.[16 17 32 33] Given the assumption of life-long immunity, age-dependent seroprevalence and risk or probability of past exposures to DENV and CHICKV, are used interchangeably.

## RESULTS

Figure 1 shows the geographical locations of households of children in the background cohort and the locations of those sampled for this analysis (151). There was an almost threefold increase in the number of reported DENV cases (42 499 cases) in 2015 compared with 2014 (15 584) and 2016 (13 612). An outbreak of CHIKV of comparable size (34 101 cases) emerged in 2015 after no previously documented history of infection in the country before 2014: Esmeraldas province accounted for one-third (10 477) of CHIKV cases in 2015 reported nationally (see online supplemental table 3 and figure 1 [showing monthly time-series of cases]). Subsequent (2016) Zika epidemic cumulative numbers are also presented for reference.

Plasma from blood samples collected between October 2015 and February 2016 from 319 individuals (including children) belonging to 151 households from 49 wards (electoral subdivisions at parish level) were analysed. Characteristics for subjects and their households are provided in table 1. Of 319 individuals, 245 (76.8%) were seropositive for DENV and 145 (45%) were seropositive for CHIKV. High numbers/proportions were positive for both arboviruses: 129 (40%) individuals belonging to 86 (57%) families living in 24 (49%) wards. Proportions of households and wards with at least one seropositive individual were: 143 (95%) households in 46 (94%) wards for DENV and 93 (62%) households in 27 (55%) wards for CHIKV.

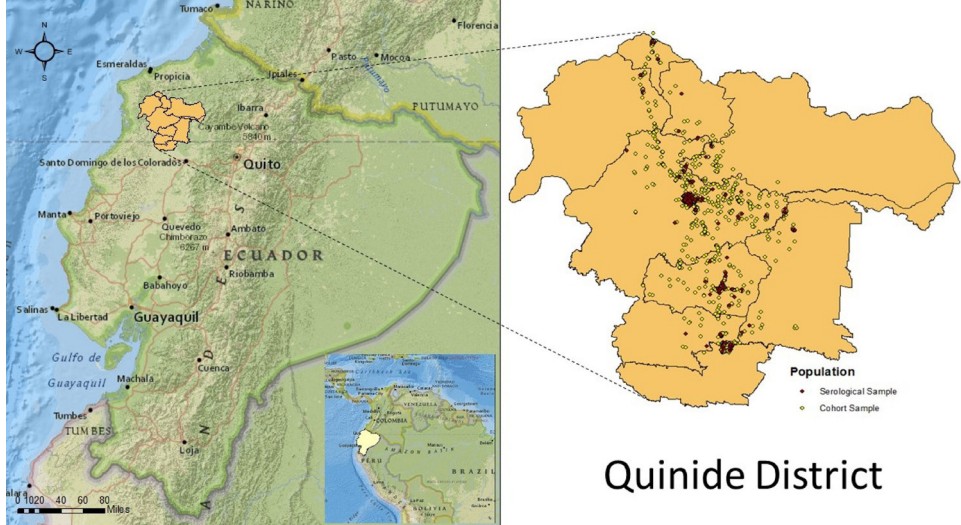

**Figure 1** Geographic distribution of the household locations of the background population (ECUAVIDA) and the serological sample.

**Table 1** Characteristics of 319 individuals from 151 households in the region of the district of Quininde, Esmeraldas Province

| Characteristic | N (%) |
|---|---|
| Individual characteristics | N=319 |
| Sex | |
| Male | 117 (37) |
| Female | 202 (63) |
| Age (years) | |
| 0–5 | 7 (2) |
| 6–10 | 140 (44) |
| 11–15 | 4 (1) |
| 16–20 | 3 (1) |
| 21–25 | 12 (4) |
| 26–30 | 45 (14) |
| 31–35 | 37 (12) |
| 36–40 | 41 (13) |
| 41–45 | 15 (5) |
| 46–50 | 9 (3) |
| >50 | 6 (2) |
| Dengue seropositive | |
| No | 74 (32.2) |
| Yes | 245 (76.8) |
| Chikungunya seropositive | |
| No | 174 (54.5) |
| Yes | 145 (45.5) |
| Household characteristics | N=151 |
| Area of residence | |
| Rural | 47 (32) |
| Urban | 104 (68) |
| Socioeconomic status | |
| Low and medium | 106 (70) |
| High | 45 (30) |
| Household dispersal (metres) | |
| <50 | 13 (8.6) |
| 51–250 | 63 (41.7) |
| 251–500 | 34 (22.5) |
| 501–1000 | 12 (8) |
| >1000 | 29 (19) |

The subsample of the cohort used in this analysis appeared to represent the overall cohort well in terms of socioeconomic status and area of residence (see online supplemental table 1). Distributions by parish and age for the census population of Esmeraldas Province, cohort and serological sample are shown in online supplemental tables 2 and 3, respectively, along with stratum-specific weights subsequently used in modelling for population-level representativeness. Parameter estimates (conditional ORs) which measure the univariate and adjusted associations of the risk of testing positive for DENV, CHIKV and both with each potential risk factor (age, sex, area of residence, socioeconomic status and distance to next door neighbour) are shown in table 2. Given the relative complexity of this non-linear modelling, the results are best understood through a predicted average seroprevalence across the population, with reference to trends and patterns by age and household characteristics (see online supplemental table 4).

Average overall DENV seroprevalence (including those seropositive for both viruses) was 84% (95% CrI: 75%–91%) but increased rapidly with age from 14% (95% CrI: 1.5%–45%) in infants to 97% (95% CrI: 93%–99%) in adults aged 60 years (figure 2). The overall average seroprevalence of CHIKV was 27% (95% CrI: 8.7%–51.6%), peaked at 42% (95% CrI: 18%–66%) in children aged 9 years and declined to 18% (95% CrI: 5%–38%) by 60 years (figure 2).

Age-dependent prevalence curves for D+C–, D–C+, D+C+ and actual proportions from raw data (bars) for 5-year age groupings are shown in figure 3. The predicted curves fitted observed data well: raw proportions of positives were generally close to the 95% CrI predicted curves. Average seroprevalence of D+C– was 58% (95% CrI: 38%–74%), that of D–C+ was 1% (95% CrI: 0%–3%) and that of D+C+ was 26% (95% CrI: 8%–50%) (see online supplemental table 5). Seroprevalence increased rapidly following a power law (ie, faster than exponential) with age for D+C–, declined rapidly for D–C+ and increased rapidly to a peak around 9 years (40%, 95% CrI: 18%–64%) for D+C+, followed by a steady decline to 18% (95% CrI: 6%–37%) by 60 years.

Average seroprevalence by groups defined by co-exposures including sex, area of residence and socioeconomic status are shown in online supplemental table 5 and their sole effects on age-specific prevalence in table 2 and online supplemental figure 2. Women were more likely to be seropositive for CHIKV (D+C+ and D–C+) but less likely to be singly positive to DENV (D+C–). The difference in D+C– seroprevalence between men and women was greatest in adolescence while that for D+C+ was greatest in childhood (see online supplemental figure 3). Individuals residing in rural areas were more likely to be D+C– while D+C+ was more frequent in urban areas; while shorter distance to nearest household (ie, household dispersal) appeared to be a risk factor for D+C+.

Strength of associations was preserved in the multivariable model but estimates were less precise as more information was extracted, that is, the CrIs were wider (final model in table 2). Men remained more likely to be D+C– and less likely to be D+C+ (see online supplemental figure 4); individuals living in rural areas were more likely to be D+C– while those living in urban areas were more likely to be D+C+ (see online supplemental figure 5); and individuals of lower economic status appeared to be more likely to have all outcomes across all ages although differences were greater among those aged less

**Table 2** Parameter estimates derived after fitting simultaneous logistic models with increasing number of explanatory variables

| Model | Dengue conditioned on chikungunya | | | | | | Chikungunya conditioned on dengue | | | | | |
|---|---|---|---|---|---|---|---|---|---|---|---|---|
| | Mean | SD | Median | P025 | P975 | P(OR<=1)/P(OR>1) | Mean | SD | Median | P025 | P975 | P(OR<=1)/P(OR>1) |
| **First models** | | | | | | | | | | | | |
| Age (power)* | 7.671 | 3.123 | 7.023 | 3.616 | 15.450 | | 0.357 | 0.097 | 0.346 | 0.202 | 0.579 | |
| **Second models** | | | | | | | | | | | | |
| Female versus male | 0.728 | 0.373 | 0.650 | 0.248 | 1.665 | **0.822/0.178** | 2.971 | 1.587 | 2.614 | 1.030 | 7.004 | **0.017/0.983** |
| Rural versus urban | 1.490 | 1.431 | 1.099 | 0.260 | 5.020 | 0.441/0.559 | 0.252 | 0.337 | 0.159 | 0.023 | 1.040 | **0.975/0.025** |
| Crowding | 1.217 | 0.744 | 1.042 | 0.348 | 3.118 | 0.476/0.524 | 0.916 | 0.759 | 0.718 | 0.179 | 2.808 | 0.679/0.321 |
| Low versus high | 3.302 | 2.472 | 2.671 | 0.828 | 9.518 | **0.052/0.948** | 0.916 | 0.858 | 0.675 | 0.142 | 3.148 | 0.684/0.316 |
| Household dispersal | 0.825 | 0.314 | 0.772 | 0.380 | 1.585 | 0.771/0.229 | 0.711 | 0.384 | 0.628 | 0.225 | 1.685 | **0.826/0.174** |
| **Final models** | | | | | | | | | | | | |
| Age (power) | 9.980 | 4.623 | 8.955 | 4.290 | 21.760 | | 0.263 | 0.085 | 0.251 | 0.132 | 0.460 | |
| Female versus male | 0.651 | 0.347 | 0.577 | 0.210 | 1.528 | **0.87/0.13** | 3.804 | 2.356 | 3.238 | 1.152 | 9.831 | **0.01/0.99** |
| Rural versus urban | | | | | | | 0.218 | 0.479 | 0.099 | 0.008 | 1.167 | **0.98/0.02** |
| Low versus high | 3.028 | 1.992 | 2.536 | 0.827 | 8.085 | **0.05/0.95** | | | | | | |
| Household dispersal | | | | | | | 0.794 | 0.561 | 0.657 | 0.186 | 2.202 | 0.76/0.24 |

*First models have age only as predictor. Second models show effects of each of age, sex, area of residence, socioeconomic status, household overcrowding and household dispersal on the OR of past exposure versus no exposure to the viruses. Final models included all variables for which the effect in the simpler models, measured as (P(OR>1) or P(OR<1)), was greater than 0.80 or less than 0.20 (shown in bold). Age-dependence is parameterised according to a power law (ie, age$^{power}$) and the corresponding estimate represented the power of the age$^{power}$ component.

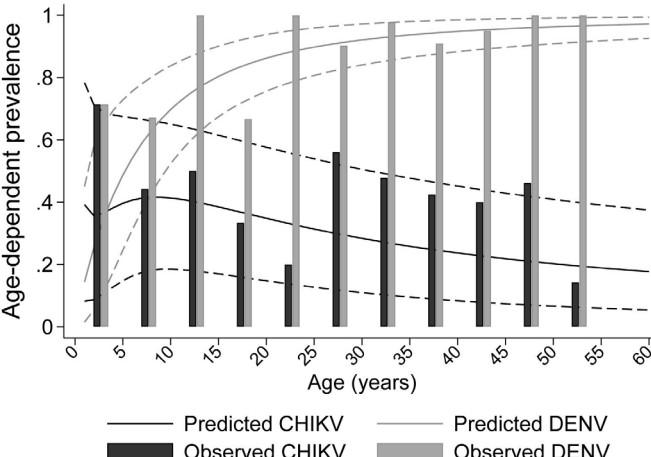

**Figure 2** Predicted age-dependent seroprevalence for dengue and chikungunya. Dashed lines represent the 95% credible intervals and the bars represent the observe proportions of seropositive participants. CHIKV, chikungunya virus; DENV, dengue virus.

than 20 years (see online supplemental figure 6). Online supplemental figures 7 and 8 show further evidence for a good fit of the model to the numbers in the data rather than to proportions: group-specific defined by age and geographical regions predicted number of seropositive individuals were derived by averaging the age-specific proportions in that group and used to predict approximate average number of exposed individuals. All the parameter estimates Markov chain Monte Carlo traces for the most complex model (table 2) and their correlations are presented in online supplemental figure 9 and 10, respectively. The estimated variance–covariance matrices corresponding to the random effects associated with both ward and family structures can also be viewed in online supplemental figure 11.

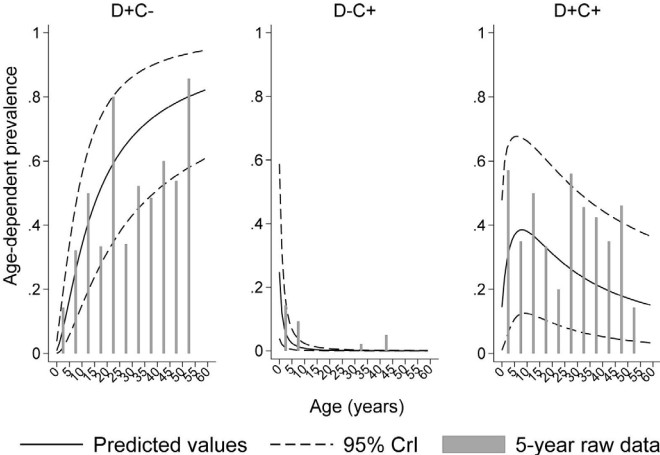

**Figure 3** Predicted age-dependent prevalence for exposure categories to dengue and chikungunya in the sample population plotted against observed proportions (in 5-year age groupings). Dashed lines represent 95% credible intervals (CrIs).

## DISCUSSION

Arbovirus infections are a growing public health threat in Latin America where DENV is endemic in tropical and subtropical regions, and CHIKV and Zika virus infections have recently emerged.[6] Despite evidence for co-circulation within the same geographical regions of more than one type of arbovirus, and in many regions all three viruses[3 9 34] that share the same vectors and cause similar symptomatology,[35] there are no studies addressing the epidemiology of co-exposures. Nevertheless, the epidemiological and statistical methodologies developed here can be made to accommodate relatively easily more than two concomitant exposures. Although DENV has been endemic in Ecuador since 1980, there are no previous seroprevalence estimates from Ecuador for either arbovirus.[6] Recent research in Machala in southern coastal Ecuador has highlighted the need for integrated surveillance[13] and predictions in the same region have been discussed in the context of El Niño Southern Oscillation climatic phenomena.[36 37] In the present study, we estimated that 84% (95% CrI: 75%–91%) of individuals aged 0–60 years had been exposed to DENV in a tropical region of coastal Ecuador. However, DENV seroprevalence increased rapidly with age reaching 97% (95% CrI: 93%–99%) by 60 years, reflecting cumulative exposures over a 30-year period since DENV re-emerged in the region in the late 1980s.[18] Placing this estimate within a global context, it concurs with a previous systematic review[6] indicating a high-average seroprevalence in the Caribbean and Latin American regions (~65%) compared with Asia (46%) and Africa (18%).

Mean chikungunya seroprevalence was 27% (95% CrI: 8.7%–51.6%) in this population and is within the range of seroprevalence estimates from seven previous studies from Latin America: Puerto Rico (23.5%), Guadeloupe (48.1%), Martinique (41.9%), Saint Martin (16.9%), Nicaragua (32.8%) and Brazil (20%).[38] A study of two urban populations in Brazil estimated CHIKV seroprevalence at 57.1% and 45.7%, respectively[38] while seroprevalence in a rural area of Chapada district (Brazil) was estimated at 20% (95% CrI: 15.4%–35%).[39] A seroprevalence of CHIKV of 75% was observed in urban children in Haiti and French Polynesia.[6] Our data showed a much flatter seroprevalence pattern by age with a peak around 9 years, likely reflecting a single intense exposure period during 2015.

In the present analysis, we developed statistical methodology to address analysis of exposures to co-circulating viruses which share strong epidemiological similarities. This method was applied to a bivariate binary outcome defined by the presence/absence of specific IgG antibodies in sera from a population sample weighted to represent the source population, and aimed at understanding how patterns of age-dependent seroprevalence vary by individual and contextual factors. The model allowed prediction of four possible outcomes (ie, D+C–, D–C+, D+C– and D–C– (used as reference group)). D+C was the most prevalent group in this population

followed by D+C+ and D−C+ (figure 3). The latter group was small and young with imprecise estimates. This statistical methodology show here can be applied relatively easily to more than two co-exposures in future studies.

Men were more likely to have been exposed to DENV while women had greater exposure to CHIKV, effects that were observed across all ages. Differences in exposure by sex could reflect temporal changes in transmission sources related to accelerating urbanisation in the context of women maintaining traditional roles as home keepers (ie, transmission is largely within and between households, particularly in an urbanising/urban setting). The observation that CHIKV seroprevalence was inversely associated with distance between households (eg, a marker of greater population densities and urbanism) suggest that CHIKV exposures are more intense in crowded places, an observation not seen for DENV in this study. These data are consistent with an outbreak in 2012 in Bangladesh that showed a reduced probability of transmission of CHIKV with increasing distance between households.[40] An increased risk of infection among women from the same study was attributed to greater time spent around the house.[40] The primary vector, *A. aegypti*, is well adapted to urban living and breeds wherever clean water is present, particularly in open recipients around households.

The non-linear age dependence of DENV seroprevalence is not unexpected given the period over which exposures are likely to have occurred. Our data indicate that by 60 years of age, almost everyone has been infected at least once with a dengue serotype. In contrast, exposure to CHIKV, which first appeared in Ecuador in 2014, possibly after a long absence,[7] was shorter: CHIKV caused a large outbreak in 2015 in coastal Ecuador, particularly in Esmeraldas Province. Thus, the age-seroprevalence profile for chikungunya, unlike dengue for which age can be viewed as a measure of length of exposure, may reflect age-dependent differences in exposures to *Aedes* vectors or susceptibility to infection.

Current global estimates of DENV and CHIKV prevalence are uncertain, being largely based on case reports to public health authorities often by other diagnostic mechanisms than laboratory analyses. Our data support the usefulness of serological surveys to inform evaluations of burden of infection within countries, particularly those with limited health resources. Further, local serological surveys will be useful for planning novel population-level interventions against arbovirus transmission such as vaccination. Dengvaxia is only currently registered dengue vaccine that might be used in endemic populations[24] and it is not licensed for use in Ecuador. There has been extensive debate over dengue vaccination planning in the endemic countries: either vaccination of populations with dengue seroprevalence rates above 80% or screening for dengue-specific antibodies and vaccinating only seropositive individuals. The WHO scientific advisory working group concluded that both 'mass vaccination based on population seroprevalence criteria' and 'pre-vaccination screening' are difficult to implement and neither can achieve herd immunity but

favours a pre-vaccination strategy in which only seropositive persons are vaccinated. A better understanding of the local epidemiology of dengue will be essential for the evaluation of cost-effectiveness of such measures.[24] Recent research suggested that highlighting areas of high prevalence is important as recent analyses suggested that transmission-blocking interventions such as *Wolbachia*, even at intermediate efficacy (50% transmission reduction), might reduce global annual disease incidence by up to 90%. The newly licensed Sanofi-Pasteur vaccine, targeting only seropositive recipients, (namely those at risk of secondary infection which can result in severe dengue and subsequent death), might reduce global annual disease incidence by 20%–30%, with the greatest impact in high-transmission settings.[25]

Limitations of seroprevalence estimates from the literature are small sample size and poorly representative samples.[6] A recent review of seroprevalence studies for DENV, CHIKV and Zika identified 185 studies of which the majority were specific populations or age sub-groups (60%), and just over one-third (38.9%) were samples from the general population[6] although few were representative samples. About half (50.3%) of the 185 studies had sample sizes of less than 500 subjects.[6] The sample size in this study likely limited our ability to detect effects of individual and contextual factors. We have minimised the potential bias in using a sample from a birth cohort rather than a general population sample, by assigning post-stratification weights based on the 2010 census.[27]

Our modelling strategy was based on assumptions that: (1) mortality caused by either infection is negligible–case fatality rate in the America region for dengue is approximately 2.5% annually while that of chikungunya is less than 0.1%[41 42]; and (2) both arboviruses induce long-lasting immunity[43]—while likely to be true for CHIKV, relevance to DENV is less clear given that immunity is strain specific and infections with four different strains are possible.[44] Molecular analysis of dengue strains in Ecuador have shown all four strains to be circulating since before 2013.[45] Further, our data can shed little light on vector competence which may affect the transmission of one or other arbovirus (or DENV serotype/genotype) if the two are circulating in the same population at the same point in time—there is some evidence for adaptation of CHIKV to *A. aegypti* and for displacement of one arbovirus by another within populations of *A. aegypti*.[46] Seroprevalence estimates here likely reflect cumulative exposures to all four DENV strains. Given relatively low-case fatality rates of 2.5% for DENV (mostly related to severe dengue) and 0.1% for CHIKV,[43 47] this assumption is reasonable. Although the serological tests used were highly sensitive, sensitivity may vary depending on whether infection is primary or secondary.[48] In the context of the present study, there was limited evidence of reduced sensitivity in older subjects (more likely to have secondary infections) among whom seroprevalence to DENV reached almost 100% by 60 years. The serological test used for CHIKV and DENV were highly specific: the study was done before the emergence of Zika virus which

first appeared in Ecuador in 2016 and in the study area in May 2016 with which serological tests for DENV share significant serological cross-reactivity.[49]

## CONCLUSIONS

Our data, from a household sample weighted to represent the general population, showed strong age-dependence of DENV seroprevalence but not CHIKV which emerged only in 2015 in the study population in a tropical region of coastal Ecuador. Our study will inform policymakers on relevant control strategies in Ecuador and decisions on cost-effectiveness of vaccination strategies for dengue.[24] Highlighting areas of high prevalence is important as recent analyses suggested that transmission-blocking interventions such as *Wolbachia*, even at intermediate efficacy (50% transmission reduction), might reduce global annual disease incidence by up to 90%.[25] The newly licensed Sanofi-Pasteur vaccine, targeting only seropositive recipients, (namely those at risk of secondary infection which can result in severe dengue and subsequent death), might reduce global annual disease incidence by 20%–30%, with the greatest impact in high-transmission settings.[25]

**Contributors** PC and ICS designed the study. PC and MC supervised data and sample collection. AL did the laboratory assays. ICS developed and implemented the statistical methodology and conducted the literature research. AR provided data and contributed to spatial data analyses. ICS and PC drafted the manuscript. NCR and JM reviewed and provided critical comments on the manuscript. All authors reviewed and approved the final version of the manuscript. ICS is the paper guarantor.

**Funding** The project was supported by the Wellcome Trust (088862/Z/09/Z) and the Universidad Internacional del Ecuador.

**Competing interests** None declared.

**Patient consent for publication** Not required.

**Provenance and peer review** Not commissioned; externally peer reviewed.

**Data availability statement** The data were from a longitudinal birth cohort study and are available upon request to pcooper@sgul.ac.uk.

**ORCID iD**
Irina Chis Ster http://orcid.org/0000-0003-2637-1259

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
