## [Reviewer comments · BMJ Open]

ARTICLE DETAILS

TITLE (PROVISIONAL)	Age-dependent seroprevalence of dengue and chikungunya: inference from a cross-sectional analysis in Esmeraldas Province in coastal Ecuador
AUTHORS	Chis Ster, Irina; Rodriguez, Alejandro; Romero, Natalia; Lopez, Andrea; Chico, Martha; Montgomery, Joel; Cooper, Philip

VERSION 1 – REVIEW

REVIEWER	Angus, Brian University of Oxford, Nuffield department of Medicine
REVIEW RETURNED	09-Jun-2020

GENERAL COMMENTS	This is a rather complex analysis of a large and unique birth cohort which proposes an interesting finding. The authors have addressed the questions from the previous reviewers and there are a large number of appendices describing the statistical approach. I would be interested in a little more detail on how the cohort were selected from the larger study but I think this is not critical to publication.
---

REVIEWER	Mary Inziani Kenya Medical Research Institute, Kenya
REVIEW RETURNED	26-Jun-2020

GENERAL COMMENTS	Age-dependent seroprevalence of dengue and chikungunya in coastal Ecuador Generic comment The article is fairly well written with a few typographical errors that should be corrected if not yet done. Page 8, Line 30 – Correct -20C to -20°C Page 12, Line 37 - Keep the terminologies consistent. D+C- instead of D+C for instance. Page 13, line 49 and 51 – Also mention how long CHIKV virus has endemic in Ecuador as well, in order to clarify what you are referring to when you say “...for either arbovirus.”
--

REVIEWER	Anna Stewart Ibarra Inter-American Institute for Global Change Research
REVIEW RETURNED	06-Jul-2020

GENERAL COMMENTS

The authors present a well written study that has potentially important data to inform future arbovirus interventions in the Americas, such as a vaccine strategy. These kinds of data are needed (and lacking) in most countries, including Ecuador, the site of this study. However, I have noted major concerns below with respect to the laboratory results, given the co-circulation of ZIKV during the study period.

Major concern: The collection of blood samples (May to Sept 2016) coincided with the peak of the ZIKV epidemic in Ecuador. Cases of ZIKV increased rapidly in northern coastal Ecuador following the major earthquake in early April 2016 (epicenter in the neighboring province of Manabi, located near the study site in Esmeraldas); however, the authors make no mention of the ZIKV epidemic ongoing during their study. 192 cases of ZIKV were reported from Esmeraldas Province in 2016, according to the Ministry of Health (see link below). In the discussion, the authors note that the first ZIKV cases were reported from the study region in May 2016. Weekly cases of ZIKA in 2016 and 2017 from the Ministry of Health are available here:

https://www.salud.gob.ec/wp-content/uploads/2015/12/vvGACETA-ZIKA_SE-38.pdf

Presumably the same populations were at risk of exposure to DENV, CHIKV and ZIKV as the viruses are transmitted by the same vector, *Aedes aegypti*. The authors do not explain how they accounted for potential cross reactivity with DENV and ZIKV in serological tests, which they recognize as a potential issue at the end of the discussion. Without testing the samples for ZIKV antibodies (or providing clear evidence that the virus was not co-circulating, which seems unlikely), it is not possible to interpret the results of the DENV analysis.

The dates presented in the manuscript are confusing.

The abstract states, "We collected data and blood samples from 319 individuals belonging to 152 households following epidemics of the infections in 2015 in Latin America." It should be clearly stated that samples were collected from May to Sept 2016. (L29-31, pg 3)

In the methods section, the authors state that samples were collected from May to September 2016 (L25, pg 8) but at the end of the discussion they state "The serological test used for CHIKV and DENV were highly specific: the study was done before the emergence of Zika virus which first appeared in Ecuador in 2016 and in the study area in May 2016 with which serological tests for DENV share significant serological cross-reactivity [46]." (L48-55, pg 17). The dates in the methods and the statement in the discussion are contradictory.

Minor comments:

L35-38, pg 6. The authors report "Asymptomatic dengue, for example, can inflate the total number of cases by 40-fold." I recommend reporting a range of asymptomatic ratios from prior studies in the region, including prior studies on DENV infections from coastal Ecuador.

	L30, pg 7. Although DENG VAXIA vaccine is available in other countries in Latin America, it should be noted that the vaccine is not licensed for use in Ecuador, the location of this study. L47, pg7. Hypotheses should be more specific and based on evidence from the literature.
--	--

VERSION 1 – AUTHOR RESPONSE

Reviewer: 1

Reviewer Name: Brian Angus

Institution and Country: Oxford University, UK Please state any competing interests or state 'None declared': none declared

This is a rather complex analysis of a large and unique birth cohort which proposes an interesting finding. The authors have addressed the questions from the previous reviewers and there are a large number of appendices describing the statistical approach. I would be interested in a little more detail on how the cohort were selected from the larger study but I think this is not critical to publication.

We would like to thank this reviewer for these very positive comments on our manuscript. To clarify the question of sample selection from the larger cohort, we have changed the Materials and Methods section/Population and study sample section as follows:

"The data consist of a representative sample of households including children (and other household members) recruited into the ECUAVIDA birth cohort between 2005 and 2009 in the town of Quininde, Esmeraldas Province [26]. The District is largely rural in a tropical region of coastal Ecuador with a population of 88,000 living below 150 m altitude with mean annual temperature of 30°C and relative humidity of 80%. Main sources of income are cultivation of palm oil and tropical fruits, and timber extraction. Children (and all other members of the household to which they belonged) due to attend the cohort clinic in Quininde for routine follow up visits between October 2015 and February 2016 were invited to participate in this study according to the dates of their scheduled attendance. This approach resulted in the selection of a representative sample of ECUAVIDA cohort households (Supplementary Table 2). As is frequent for such ad-hoc surveys, post-stratification weightings [27] using the census population distributions were used to adjust the estimates to population-generalizable values (details in supplementary information and weights shown in Supplementary Table 3)."

*Supplementary Table 2 shows that the sample was representative of the entire cohort. The text of Results states, "The sub-sample of the cohort used in this analysis appeared to represent the overall cohort well in terms of socioeconomic status and area of residence (**Error! Reference source not found.**)"*

Reviewer: 2

Reviewer Name: Mary Inziani

Institution and Country: Kenya Medical Research Institute, Kenya Please state any competing interests or state 'None declared': None declared

Accept but the minor typographical revisions should be done. In view of the complex statistical calculations involved, the paper requires a specialist statistical review, if it has not yet been done.

We would like to thank this reviewer for the positive view on our manuscript. Minor typographical errors have been corrected as detailed below.

Specific comments

The article is fairly well written with a few typographical errors that should be corrected if not yet done.

Page 8, Line 30 – Correct -20C to -20°C

We have corrected this – thank you.

Page 12, Line 37 - Keep the terminologies consistent. D+C- instead of D+C for instance.

We have also corrected this – thank you.

Page 13, line 49 and 51 – Also mention how long CHIKV virus has endemic in Ecuador as well, in order to clarify what you are referring to when you say “...for either arbovirus.”

Chikungunya appeared in Ecuador in 2015 as detailed in the manuscript (see excerpts from the text below). Since 2015, the infection has declined rapidly in incidence with 10 reported cases nationally since 2018. We have also extended Supplementary Table 1 with Zika using the information in the link this reviewer generously indicated.

Introduction:

“Over the past 15 years, the number of dengue cases reported in Latin America has increased dramatically [5] while chikungunya, which emerged in the Caribbean region in 2013, has spread rapidly through the region [6] causing an estimated 2.9 million cases [7-9].”

.....

“Chikungunya first appeared in the Caribbean region in 2013, reaching Ecuador in 2014 causing a large outbreak the following year [16] [17]”

Discussion:

“In contrast, exposure to CHIKV, which first appeared in Ecuador in 2014, possibly after a long absence [7], was shorter: CHIKV caused a large outbreak in 2015 in coastal Ecuador, particularly in Esmeraldas Province. Thus, the age-seroprevalence profile for chikungunya, unlike dengue for which age can be viewed as a measure of length of exposure, may reflect age-dependent differences in exposures to Aedes vectors or susceptibility to infection.”

Reviewer: 3

Reviewer Name: Anna Stewart Ibarra

Institution and Country: Inter-American Institute for Global Change Research, United States Please state any competing interests or state ‘None declared’: None declared

The authors present a well written study that has potentially important data to inform future arbovirus interventions in the Americas, such as a vaccine strategy. These kinds of data are needed (and lacking) in most countries, including Ecuador, the site of this study.

We would like to thank this reviewer for the general positive view on our manuscript. Please find below clarifications to this reviewer’s remarks.

However, I have noted major concerns below with respect to the laboratory results, given the co-circulation of ZIKV during the study period.

Major concern: The collection of blood samples (May to Sept 2016) coincided with the peak of the ZIKV epidemic in Ecuador. Cases of ZIKV increased rapidly in northern coastal Ecuador following the major earthquake in early April 2016 (epicenter in the neighboring province of Manabi, located near the study site in Esmeraldas); however, the authors make no mention of the ZIKV epidemic ongoing during their study. 192 cases of ZIKV were reported from Esmeraldas Province in 2016, according to the Ministry of Health (see link below). In the discussion, the authors note that the first ZIKV cases were reported from the study region in May 2016. Weekly cases of ZIKA in 2016 and 2017 from the Ministry of Health are available here:

https://www.salud.gob.ec/wp-content/uploads/2015/12/vvGACETA-ZIKA_SE-38.pdf

Presumably the same populations were at risk of exposure to DENV, CHIKV and ZIKV as the viruses are transmitted by the same vector, Aedes aegypti. The authors do not explain how they accounted for potential cross reactivity with DENV and ZIKV in serological tests, which they recognize as a

potential issue at the end of the discussion. Without testing the samples for ZIKV antibodies (or providing clear evidence that the virus was not co-circulating, which seems unlikely), it is not possible to interpret the results of the DENV analysis.

The dates presented in the manuscript are confusing.

The abstract states, “We collected data and blood samples from 319 individuals belonging to 152 households following epidemics of the infections in 2015 in Latin America.” It should be clearly stated that samples were collected from May to Sept 2016. (L29-31, pg 3)

*We thank the reviewer for pointing out this inconsistency. Sample collection was done between **October 2015 and February 2016** (i.e. before the appearance of Zika virus infections in the study area from approximately May 2016 as stated in the Discussion). Sample collection dates have been corrected in the text of the revised manuscript in Material and Methods section under description of Population and study sample.*

Moreover, Supplementary Figure 1 in the manuscript clearly indicates that the two epidemics had finished their course by week 36 in Esmeraldas – before the data collection started in October (around week 40).

In the methods section, the authors state that samples were collected from May to September 2016 (L25, pg 8) but at the end of the discussion they state “The serological test used for CHIKV and DENV were highly specific: the study was done before the emergence of Zika virus which first appeared in Ecuador in 2016 and in the study area in May 2016 with which serological tests for DENV share significant serological cross-reactivity [46].” (L48-55, pg 17). The dates in the methods and the statement in the discussion are contradictory.

We apologize for the confusion and we hope we have clarified the issue – very many thanks for highlighting this important inconsistency.

Minor comments:

L35-38, pg 6. The authors report “Asymptomatic dengue, for example, can inflate the total number of cases by 40-fold.” I recommend reporting a range of asymptomatic ratios from prior studies in the region, including prior studies on DENV infections from coastal Ecuador.

We thank the reviewer for this suggestion. We have now added in the paragraph indicated above a valuable newly published reference, namely:

Melissa Vitale, Christina D. Lupone, Aileen Kenneson-Adams, Robinson Jaramillo Ochoa, Tania Ordoñez, Efraín Beltran-Ayala, Timothy P. Endy, Paula F. Rosenbaum & Anna M. Stewart-Ibarra, A comparison of passive surveillance and active cluster-based surveillance for dengue fever in southern coastal Ecuador. BMC Public Health, 2020. 20(1): p. 1065.

L30, pg 7. Although DENVVAXIA vaccine is available in other countries in Latin America, it should be noted that the vaccine is not licensed for use in Ecuador, the location of this study.

We thank this reviewer for pointing this out, we have now modified the text which refers to DENVVAXIA. More specifically, the sentence in the Discussion reads as below:

“Dengvaxia is only currently registered dengue vaccine that might be used in endemic populations [24] and it is not licensed currently for use in Ecuador.”

L47, pg7. Hypotheses should be more specific and based on evidence from the literature.

We thank this reviewer for this suggestion. We have updated our References list with recent valuable and informative work – apologies for these omissions as the publications were not available at the time of our original submission.

Petrova D, R.X., Sippy R, Ballester J, Mejia R, Beltran-Ayala E, Borbor-Cordova MJ, Vallejo GM, Olmedo AA, Stewart-Ibarra AM, Lowe R. , *he 2018-2019 weak El Niño: predicting the risk of a dengue outbreak in Machala, Ecuador*. International Journal of Climatology, 2020. **In Press**.

Petrova, D., et al., *Sensitivity of large dengue epidemics in Ecuador to long-lead predictions of El Niño*. Climate Services, 2019. **15**: p. 100096.

Our Discussion is richer and more informative now and it reads as below:

“Although DENV has been endemic in Ecuador since 1980, there are no previous age related seroprevalence estimates from Ecuador for either arbovirus [6]. Recent research in Machala in southern coastal Ecuador has highlighted the need for integrated surveillance [13] and predictions in the same region have been discussed in the context of El Niño Southern Oscillation climatic phenomena [36, 37]”

VERSION 2 – REVIEW

REVIEWER	Anna Stewart Ibarra Inter-American Institute for Global Change Research Uruguay
REVIEW RETURNED	06-Aug-2020
GENERAL COMMENTS	The authors present a well written study on DENV and CHIKV age stratified seroprevalence, providing some of the first data of this nature from Ecuador.

VERSION 2 – AUTHOR RESPONSE

We would like to thank you for this very positive view on the latest version of our manuscript.

We have changed the Patient and Public Involvement which reads as below. At the time of planning and data collection - this issue had not had such prominence as quite rightfully has nowadays.

"Patients were not involved in design or conduct of this study and there are no plans to disseminate the results to study participants."

Attached are the latest both marked and clean versions of the manuscripts.